An examination of the Devonian fishes of Michigan

Stack Jack stackj@sas.upenn.edu
Sallan Lauren lsallan@upenn.edu
Department of Earth and Environmental Science, University of Pennsylvania , Philadelphia , PA , United States of America
Sues Hans-Dieter
Electronic publication date: 2018 Sep 20
Publication date: 2018
Volume: 6
Electronic Location ID: e5636
Received 2018 May 29; Accepted 2018 Aug 24
Copyright: ©2018 Stack and Sallan
Copyright year: 2018
Copyright holder: Stack and Sallan
License: This is an open access article distributed under the terms of the Creative Commons Attribution License, which permits unrestricted use, distribution, reproduction and adaptation in any medium and for any purpose provided that it is properly attributed. For attribution, the original author(s), title, publication source (PeerJ) and either DOI or URL of the article must be cited.
License URL: https://creativecommons.org/licenses/by/4.0/

Keywords: Paleoichthyology, Fossil, Paleontology, Michigan basin, Placoderm, Vertebrate, Reefs, Appalachian basin, Cleveland shale, Ichthyology

Funding: University of Pennsylvania This work was supported by the Paleontological Society Rodney M. Feldmann Award (to Jack Stack), the University of Pennsylvania Paleontology Summer Stipend (to Jack Stack), and the University of Pennsylvania Grant for Faculty Mentoring Undergraduate Research (to Lauren Sallan). The funders had no role in study design, data collection and analysis, decision to publish, or preparation of the manuscript.

==============================
We surveyed the taxa, ecosystems, and localities of the Devonian fishes of Michigan to provide a framework for renewed study, to learn about the diversity and number of these fishes, and to investigate their connection to other North American faunas. Nineteen genera of fishes have been found in the Middle and Late Devonian deposits of Michigan, of which thirteen are ‘placoderms’ represented by material ranging from articulated head shields to ichthyoliths. As expected from the marine nature of these deposits, ‘placoderms’ are overwhelmingly arthrodire in nature, but two genera of ptyctodonts have been reported along with less common petalichthyid material. The remaining fish fauna consists of fin-spines attributed to ‘acanthodians’, two genera of potential crown chondrichthyans, an isolated dipnoan, and onychodont teeth/jaw material. There was an apparent drop in fish diversity and fossil abundance between Middle and Late Devonian sediments. This pattern may be attributed to a paucity of Late Devonian sites, along with a relative lack of recent collection efforts at existing outcrops. It may also be due to a shift towards open water pelagic environments at Late Devonian localities, as opposed to the nearshore reef fauna preserved in the more numerous Middle Devonian localities. The Middle Devonian vertebrate fauna in Michigan shows strong connections with same-age assemblages from Ohio and New York. Finally, we document the presence of partially articulated vertebrate remains associated with benthic invertebrates, an uncommon occurrence in Devonian strata outside of North America. We anticipate this new survey will guide future field work efforts in an undersampled yet highly accessible region that preserves an abundance of fishes from a critical interval in marine vertebrate evolution.

Introduction

The Devonian Period (419.2–358.9 Ma), the so-called “Age of Fishes,” was marked by major transitions in vertebrate biodiversity, including the takeover of ecosystems by jawed fishes, the first appearance of tetrapods, and several group-specific and global extinctions (Friedman & Sallan, 2012; Ogg, Ogg & Gradstein, 2016). Despite the evolutionary importance of this interval, Devonian vertebrates from the US have been undersampled and understudied in the last 50 years relative to specimens of the same age from the UK, China, Australia and even Antarctica (Long, 1995), with possible exceptions being select very Late Devonian faunas such as the Cleveland Shale and Red Hill (Elliott et al., 2000; Carr & Jackson, 2008; Daeschler & Cressler, 2011). Fishes are common in fossil-bearing Paleozoic strata throughout the midwestern US, with many outcrops discovered over a century ago (Newberry, 1889; Eastman, 1907; Eastman, 1908; Elliott et al., 2000). The Devonian-aged vertebrate fauna from Michigan is abundant, but is poorly documented in the scientific literature relative to similar strata in Ohio, Pennsylvania and elsewhere, despite heavy, ongoing collection efforts by amateurs (Elliott et al., 2000). We undertook this survey of Devonian fishes from Michigan, the first since 1970, as a result of new discoveries by J.S. (with the assistance of avocational paleontologists).

During the Devonian, the Michigan Basin was located between, and connected to, the better described Illinois (Hussakof, 1913; Cluff, 1980; Brusatte, 2007) and Appalachian Basins (Newberry, 1873; Claypole, 1883; Carr & Jackson, 2008; Carr & Hlavin, 2010; Downs, Criswell & Daeschler, 2011; Daeschler & Cressler, 2011), yet it has been largely ignored by researchers for over a century. Most of the handful of studies on Devonian vertebrates from Michigan are descriptions of single taxa (e.g., Stevens, 1964; Miles, 1966; Schultze, 1982; Carr & Jackson, 2005). Only one study from the last century attempted to survey Devonian fossil fishes from Michigan, but was limited in scope to arthrodires (Case, 1931). Several decades later, an additional summary of vertebrate fossils from the University of Michigan Museum of Paleontology was published by Dorr & Eschman (1970), accompanied by a description of the Devonian ecosystems from Michigan, including information on collecting sites, geologic history, and invertebrate faunae. However, many of their identifications were incorrect, and some of their figured and described specimens are currently missing from the UMMP collections.

Recent fieldwork in Michigan, undertaken mostly by amateur collectors, has revealed a diverse and in many ways distinct Devonian vertebrate fauna containing taxa not reported by Case (1931) and Dorr & Eschman (1970). New sites have produced an abundance of relatively well-preserved vertebrate skeletal remains, including articulated material. Of the 201 Devonian vertebrate specimens from Michigan that are catalogued in museums, 118 were collected by J.S. after the previous survey. Here, we provide an updated and comprehensive summary of what is known about the Devonian fish fauna from Michigan. We also compare the Devonian fish fauna of Michigan to similarly aged marine faunas from New York and Ohio, placing it within the larger regional context of the Devonian of North America.

Materials and Methods

We surveyed Michigan Devonian fish specimens in the collections of the University of Michigan (UMMP), the Michigan History Museum (BV, JS, M, or a date), Michigan State University Museum (VP), the Cleveland Museum of Natural History (CMNH), and the literature, which contains records for other specimens from the Michigan State University Museum, the Cleveland Museum of Natural History, and the Great Lakes Area Paleontological Museum (GLAPM). We also performed surveys of known outcrops as described in text. Collections from localities on public lands were approved by the State of Michigan, State Historic Preservation Office Permit AE2016-10, and collections were deposited in the Michigan History Museum. We compiled and organized information including, specimen counts, geologic setting and localities (full list of localities available in supplemental files). A full list of examined specimens is available as a supplement. Below, we describe the occurrence and distribution of vertebrate remains by formation alongside information on associated invertebrate remains, depositional environment, and the locations of vertebrate-bearing fossil sites. We also summarize the characteristics of the Devonian vertebrates resident in Michigan. We then synthesized temporal patterns, ecology, and faunal similarities with other marine Devonian localities.

Middle Devonian Geological Distribution

Pre-Traverse Group

The oldest vertebrate bearing Devonian localities in Michigan are from the Eifelian (Middle Devonian; 393.3–387.7 Ma) (Swezey, 2002; Brett et al., 2011; Ogg, Ogg & Gradstein, 2016; Figs. 1 and 2). These occur in the Dundee Limestone formation, which has also produced numerous invertebrate remains (Dorr & Eschman, 1970; Ehlers & Kesling, 1970). The fish found in the Dundee Limestone include the onychodont Onychodus sigmoides Newberry, 1873 and Onychodus sp., the stem-chondrichthyan ’acanthodian’ Machaeracanthus sp. Newberry, 1873, the chondrichthyan Acondylacanthus gracillimus St. John & Worthen, 1875, the presumptive arthrodire ?Titanichthys sp. Newberry, 1885, the ptyctodont Ptyctodus sp. Pander, 1858, and the petalichthyid ?Macropetalichthys sp. Norwood & Owen, 1846 (Dorr & Eschman, 1970; Fig. 3). ?Titanichthys sp. (UMMP 26114), A. gracillimus (UMMP 26523), and Machaeracanthus sp. (UMMP 26111, UMMP 26112) and a jaw (UMMP 26113) from Onychodus sp. are known from Sibley Quarry, Wyandotte, Wayne County (Dorr & Eschman, 1970; Table 1). An isolated fin spine from Machaeracanthus sp. (UMMP 3521) and an isolated tooth from O. sigmoides (UMMP 22006; Fig. 3A) are documented from a site in London Township, Monroe County (Dorr & Eschman, 1970). One specimen of two articulated armor plates (UMMP 14320; Fig. 3B) from ?Macropetalichthys sp. and another specimen of Ptyctodus sp. (UMMP 14321) are documented from a locality near Trenton, in Wayne County (Dorr & Eschman, 1970). It is important to note that the geological source of the Ptyctodus specimen is uncertain and cannot be verified based on the matrix (Dorr & Eschman, 1970). Despite this taxonomic diversity, vertebrate abundance and perhaps preservation potential within the Dundee Limestone appears to be very poor (Ehlers & Kesling, 1970).

Figure 1 Stratigraphy of the Devonian deposits of the northern part of the Lower Peninsula of Michigan.

Figure modified from Elliott et al. (2000), Fig. 3.

Figure 2 Chart showing the correlation between international and North American Devonian stage names.

Figure modified from Swezey (2002). US Geological Survey.

Figure 3 Vertebrate remains from the Dundee Limestone Formation.

(A) a large tooth from O. sigmoides, from the Dundee Limestone of London Township, Monroe County, UMMP 22006. Scale bar equals 1 cm. (B) a spinal and anterior ventrolateral plate from ?Macropetalichthys sp. (previously identified as Arctolepis sp.), from the Dundee Limestone near Trenton, UMMP 14320. Abbreviations: Sp, spinal; Spi, spines of the spinal plate; Avl, anterior ventrolateral. Scale bar equals 1 cm.

The Traverse Group

The Traverse Group encompasses all but two of the known vertebrate-bearing formations from Michigan (Dorr & Eschman, 1970; Fig. 1). It was deposited in the Givetian (387–382.7 Ma), or the Erian (391.8–388 Ma) regional series (Swezey, 2002; Gradstein, Ogg & Smith, 2004; Ogg, Ogg & Gradstein, 2016; Fig. 2). Six separate depositional environments or zones have been sampled from the Traverse group, representing different water depths: a lagoonal zone, the zone of turbulence, the stromatoporid-coral zone, the coral-brachiopod zone, the diverse fauna zone, and a bioherm (Ehlers & Kesling, 1970). These zones were identified and described by Ehlers & Kesling (1970), and are briefly summarized here for future reference. The stromatoporid-coral zone was nearshore, shallow, and contained invertebrates such as brachiopods and crinoids (Ehlers & Kesling, 1970). The coral-brachiopod zone represents deeper water coincident with the lowest limit of stromatoporoids, with fossil material consisting mostly of brachiopods, corals, and bryozoans (Ehlers & Kesling, 1970). Rocks from both of these zones are abundant in the Traverse Group, and tend to be medium to fine grained limestones that can grade down into calcareous shales (Ehlers & Kesling, 1970). The diverse fauna zone was reefal with abundant vertebrates, brachiopods, trilobites, and crinoids with less common corals, bryozoans, and mollusks (Ehlers & Kesling, 1970). The rocks from the diverse fauna zone tend to be thick claystones or shale beds, with low calcareous content (Ehlers & Kesling, 1970). It is also possible to find fish and invertebrate fossils in the lagoonal sediments, but these may have been the result of marine incursions rather than distinct faunas (Stevens, 1964; Ehlers & Kesling, 1970). The rocks from the lagoonal zone are lithographic limestone (Ehlers & Kesling, 1970).

Vertebrate distribution in Early Erian (Givetian) deposits

Bell Shale

One tooth plate (UMMP 14460) from Ptyctodus sp. has been reported from the Bell Shale of Rogers City, Presque Isle County (Dorr & Eschman, 1970). Invertebrate material suggests that the Bell Shale was deposited in the diverse fauna zone (Pohl, 1930; Ehlers & Kesling, 1970).

Rockport Quarry Limestone

The fishes found in the Rockport Quarry Limestone include the arthrodires Protitanicthys rockportensis Case, 1931, ?Holonema rugosum Claypole, 1883, Holonema sp. Newberry, 1889, Dunkleosteus sp. Lehman, 1956, Mylostoma sp. Newberry, 1883 and Dinomylostoma sp. Eastman, 1906, the ptyctodont Ptyctodus sp., the ’acanthodian’ ?Machaeracanthus sp., and the chondrichthyan ?Tamiobatis sp. Eastman, 1897a; (Dorr & Eschman, 1970; J Stack, pers. obs., 2018; Fig. 4). There are also two specimens (7M, VP. 522) of placoderms of unknown affinity that have been collected from the Rockport Quarry Limestone. One of these specimens (7M) consists of armor fragments distinguishable from other resident taxa by a lack of tubercles (Fig. 4B), but further material is required to make an exact attribution. Also, a single specimen (UMMP 3898) was given an uncertain designation as ?Holonema rugosum by Dorr & Eschman (1970).

Figure 4 Vertebrate remains from the Rockport Quarry Limestone Formation.

(A) a partial skull roof from Holonema sp., from the Rockport Quarry Limestone at the abandoned Kelly Island Limestone Quarry at Rockport State Park, Alpena County, UMMP 12991. Abbreviations: Nu, Nuchal; PNu, Paranuchal; C, Central Plate. Scale bar equals 2 cm. (B) The remains of an unidentified placoderm from the Rockport Quarry Limestone at the abandoned Kelly Island Limestone Quarry at Rockport State Park, Alpena County, 7M, Michigan History Museum. Scale bar equals 1 cm. (C) an incomplete right anterior ventrolateral from Dunkleosteus sp., from the Rockport Quarry Limestone of Rockport Quarry, Alpena County, UMMP 16156. Scale bar equals 2 cm. (D) a small spine from ?Tamiobatis sp. (previously identified as Ctenacanthus sp.) from the Rockport Quarry Limestone at the abandoned Kelly Island Limestone Quarry at Rockport State Park, Alpena County, UMMP 13147. Scale bar equals 1 cm.

The Rockport Quarry Limestone contains the most diverse Devonian vertebrate fauna in Michigan (Dorr & Eschman, 1970; Sallan & Coates, 2010). The main vertebrate-bearing outcrop is at the abandoned Kelly Island Limestone Quarry at Rockport State Park, Alpena County (Dorr & Eschman, 1970). The degree of preservation is often good, with partially articulated armor plates frequently observed in the field, but mining operations have damaged many of the accessible fossils (J Stack, pers. obs., 2016). Fortunately, this locality is the most productive Devonian vertebrate site in Michigan by far (J Stack, pers. obs., 2018), and better specimens are likely to be recovered in the future. Despite the relatively high diversity and abundance of fish material and collection efforts by amateurs, little material has been accessioned in state museums. The UMMP contains only single specimens of ?Machaeracanthus sp. (UMMP 13047), Dunkleosteus sp. (UMMP 16152; Fig. 4C), Mylostoma sp. (UMMP 13612), Ptyctodus sp. (UMMP 13045), and ?Tamiobatis sp. (UMMP 13147). There are 59 specimens of P. rockportensis and 26 specimens of Holonema sp. deposited in UMMP, MSU and the MHM, and many more recently recovered fossils of those fishes reside in private collections (J Stack, pers. obs., 2016). The large numbers of stromatoporoids and corals, alongside less common brachiopods, trilobites, and crinoids, suggests that the vertebrate bearing rocks of the Rockport Quarry Limestone were deposited in the stromatoporoid-coral zone (Ehlers & Kesling, 1970). In contrast to most Devonian sites, articulated vertebrate remains are often preserved in conglomerates with invertebrate specimens (J Stack, pers. obs., 2016). It is notable that the abundance and diversity of coincident invertebrates is considerably lower than the underlying Bell Shale, most likely because of the difference in depositional zone (Ehlers & Kesling, 1970).

Genshaw Formation

A single specimen of an incomplete armor plate (UMMP 4169) from the ptyctodont ?Eczematolepis sp. Miller, 1892 has been documented from the Genshaw Formation near Posen, Presque Isle County (Dorr & Eschman, 1970). In addition, a specimen of ?H. rugosum (UMMP 3899) has been reported from the Killians member of the Genshaw Formation, at a locality referred to as French Road near Long Lake, near Rockport Quarry, Alpena County (Dorr & Eschman, 1970). Invertebrates found in this formation are typical of the diverse fauna zone (Ehlers & Kesling, 1970).

Vertebrate distribution in Middle Erian (Givetian) deposits

Newton Creek Limestone

A single specimen of the lungfish (dipnoi) Chirodipterus onawayensis (unnumbered) Schultze, 1982, along with two specimens of Machaeracanthus sp. (UMMP 47691 and UMMP 47692) and two specimens of Holonema farrowi Stevens, 1964 (UMMP 46647, UMMP 46648) are documented from the Newton Creek Limestone (Stevens, 1964; Dorr & Eschman, 1970; Schultze, 1982). These fossils were collected from the north edge of the Onaway Stone Quarry, Presque Isle County, where the Newton Creek Limestone is referred to as the Koehler Limestone (Dorr & Eschman, 1970; Ehlers & Kesling, 1970). While these fishes were recovered from the lagoonal zone (Stevens, 1964), some may have been deposited during a deeper marine incursion (Ehlers & Kesling, 1970).

Gravel Point Formation

This formation has produced single catalogued specimens of Gyracanthus sp. Woodward, 1906 (UMMP 1329) ?Onychodus sp. (UMMP 14370) (Dorr & Eschman, 1970), and a holonemiid (UMMP 3129) (Dorr & Eschman, 1970). These specimens were found at South Point (Gravel Point), Little Traverse Bay, Charlevoix County (Ehlers & Kesling, 1970). The invertebrate fossils from this formation are typical of the diverse fauna zone (Pohl, 1930; Ehlers & Kesling, 1970).

Alpena Limestone Formation

This formation was deposited contemporaneously with the Gravel Point Formation (Ehlers & Kesling, 1970). Single specimens of Dunkleosteus sp. (UMMP 16152) and Ptyctodus sp. (UMMP 16157), along with three specimens of ?Mylostoma (BV3, BV6, and BV7) and a single specimen of some partially articulated pieces of armor from the head shield of ?Macropetalichthys sp. (BV 4) comprise the catalogued vertebrate material from the Alpena Limestone Formation (Dorr & Eschman, 1970). The specimens of Dunkleosteus sp. and Ptyctodus sp. are from a locality referred to as Alkali Quarry in Alpena, and the ?Mylostoma sp. and ?Macropetalichthys sp. specimens are from the Besser Museum Fossil Park in Alpena (J Stack, pers. obs., 2018; Dorr & Eschman, 1970). Amateur collectors have reported P. rockportensis and other vertebrates from this formation, but none of these specimens are deposited in museum collections. The invertebrate fossils from this formation are typical of the coral-brachiopod zone (Pohl, 1930; Ehlers & Kesling, 1970). Many of these are articulated and occur in conglomerates with similarly partially articulated vertebrate specimens.

Vertebrate distribution in Late Erian (Givetian) deposits

Four Mile Dam Formation

?Mylostoma sp., P. rockportensis, ?Macropetalichthys sp. and an unidentified ’acanthodian’ and ’placoderm’ have recently been collected by one of the authors (JS) from the Four Mile Dam Formation (Fig. 5). At this time, ?Mylostoma sp. is known from 39 specimens, while an unidentified ’placoderm’ is known from one specimen (JS 4; Fig. 5C), an unidentified ’acanthodian’ is known from two partial fin spines (JS 120, JS 121; Fig. 5A), P. rockportensis is known from one specimen of an armor fragment (JS 101), and ?Macropetalichthys sp. is known from a partial head shield (32M). The unidentified placoderm (JS 4) is a specimen of an armor plate that does not resemble the other resident placoderms (?Mylostoma sp., P. rockportensis, and ?Macropetalichthys sp., Fig. 5C). More material needs to be collected before a concrete identification is made. Likewise, the two isolated acanthodian fin spines are distinct from named acanthodians (Gyracanthus, Machaeracanthus, and Oracanthus) and chondrichthyans (Tamiobatis and Acondylacanthus) reported from Michigan (J Stack, pers. obs., 2018; Denison, 1979; Maisey, 1982; Williams, 1998). The vertebrate fauna from the Four Mile Dam Formation has high abundance but low diversity compared to the Rockport Quarry Limestone (Dorr & Eschman, 1970). The main vertebrate-bearing outcrop of this formation is the Specific Stone Products Quarry in Alpena, with the fossils recovered from discarded piles of limestone from this quarry on the shores of Betsie Bay in Elberta, Benzie County (J Stack, pers. obs., 2018).

Figure 5 Vertebrate remains from the Four Mile Dam Formation.

(A) a broken spine from an unidentified acanthodian, from the Four Mile Dam Formation of the Betsie Bay Rockpiles, Elberta, Benzie County, JS 121, Michigan History Museum. Scale bar equals 0.5 cm. (B) a flattened specimen of a trunk shield from ?Mylostoma sp., from the Four Mile Dam Formation of the Betsie Bay Rockpiles, Elberta, Benzie County, 21M. Abbreviations: MD, Median dorsal; ADL, Anterior dorsolateral; Nu, Nuchal. Scale bar equals 1 cm. (C) a partial armor plate from an unidentified placoderm, from the Four Mile Dam Formation of the Betsie Bay Rockpiles, Elberta, Benzie County, JS 4, Michigan History Museum. Scale bar equals 1 cm. (D) a piece of limestone containing both crinoid heads and an armor plate from ?Mylostoma sp. from the Four Mile Dam Formation of the Betsie Bay Rockpiles, Elberta, Benzie County, 9M, Michigan History Museum. This specimen is an example of the association between vertebrates and invertebrates in the Middle Devonian deposits of Michigan. The solid arrow indicates the armor plate and the dashed arrow indicates a crinoid calyx. Scale bar equals 1 cm.

Despite the fact that the Four Mile Dam Formation was only recently identified as a vertebrate-bearing formation, fish material may be more abundant in these sediments than at any other Devonian locality in Michigan. There have been 44 vertebrate specimens recovered in a few years of limited yet deliberate collecting in rocks from this formation. However, this represents more recent collection effort by university-affiliated researchers than has recently been expended at other Michigan sites. Many of the Four Mile Dam ?Mylostoma sp. specimens are well-preserved and partially articulated, including potential juveniles, suggesting that this locality holds high potential for future research (J Stack, pers. obs., 2018). The degree of preservation suggests rapid burial, perhaps by a mudflow initiated by a storm given the environmental setting. This interpretation is supported by the fact that many of the vertebrate specimens from this formation are found in conglomerates with invertebrate specimens (Fig. 5D). This includes fully three-dimensional, articulated crinoid calyces, a rarity in the Devonian record that requires quick deposition (Fig. 5D). These invertebrates are found in other outcrops of the Four Mile Dam Formation, but fish fossils are absent from all but the site mentioned above (Dorr & Eschman, 1970; Ehlers & Kesling, 1970), implying that vertebrate material may not be preserved in this formation under more normal environmental conditions. Invertebrate fossils found in this formation are typical of the diverse fauna zone (Ehlers & Kesling, 1970).

Norway Point Formation

A single spine (UMMP 23495), attributed to the acanthodian Oracanthus sp. Agassiz, 1833–1844, has been reported from the Norway Point formation at the Four Mile Dam, 6 kilometers northwest of Alpena, Alpena County (Ehlers & Kesling, 1970; Dorr & Eschman, 1970). The invertebrates from this formation suggest that the rocks were deposited in the coral-brachiopod zone (Ehlers & Kesling, 1970).

Potter Farm Formation

One specimen of Ptyctodus sp. (UMMP 21718) is known from the Potter Farm Formation, recovered from a locality referred to by Dorr & Eschman (1970) as “Old Wamer’s Brickyard”, southwest of Alpena, Alpena County. However, Dorr & Eschman (1970) noted that the exact geologic affinity of this specimen is uncertain (it may not have been collected in situ). However, another specimen of Ptyctodus sp. (UMMP 21817) was collected from an outcrop of the Potter Farm Formation at the ”west edge of Alpena Cemetery”, Alpena (Dorr & Eschman, 1970). Amateur collectors have also reported specimens of P. rockportensis from this formation, but these specimens are not deposited in museum collections and therefore cannot be verified. The invertebrate fossils found from the Potter Farm Formation are typical of the diverse fauna zone (Ehlers & Kesling, 1970). Many of these invertebrates are again preserved in conglomerates with vertebrates, as in the Four Mile Dam Formation (Fig. 5D). These invertebrate specimens are occasionally broken and preserved in a way that superficially resembles dark, thin armor plates from vertebrates, and have been misidentified as such by amateurs, so caution must be used when identifying specimens from this formation without attention to histology.

Thunder Bay Limestone

A tooth plate (UMMP 3023) from Ptyctodus sp. has been reported from the Thunder Bay Limestone (Dorr & Eschman, 1970). This specimen was collected from the bluffs on the northeast shore of Partridge Point, 6.4 kilometers south of Alpena (Dorr & Eschman, 1970). The invertebrates from this formation are typical of the diverse fauna zone (Pohl, 1930; Ehlers & Kesling, 1970).

Late Devonian Geological Distribution

Antrim Shale

The Antrim Shale was deposited in the Frasnian (382.7–372.2 Ma), or Senecan (388–370 Ma) regional series and in the Famennian (372.2–358.9 Ma), or the Chatauquan (370–359.2 Ma) regional series of the Late Devonian (Dorr & Eschman, 1970; Ehlers & Kesling, 1970; Gutschick & Sandberg, 1991; Gradstein, Ogg & Smith, 2004; Ogg, Ogg & Gradstein, 2016; Fig. 1; Fig. 2). Fish from the Antrim Shale that are accessioned in museum collections include Diplognathus lafargei Carr & Jackson, 2005, ?Trachosteus clarki Newberry, 1889, Dunkleosteus sp., and Aspidichthys clavatus Newberry, 1873, all known only from single specimens (Dorr & Eschman, 1970; Carr & Jackson, 2005). There are also reports of the presence of ptyctodonts, cladodonts (Chondrichthyes), conodonts, and ray-finned fishes from the Antrim Shale (Ehlers & Kesling, 1970; Elliott et al., 2000). It is not clear if these specimens are deposited in museum collections, and therefore they were not examined in this study. ?T. clarki is known from an isolated inferognathal (UMMP 18206) from a locality 1.6 kilometers north of Norwood, Charleviox County (Dorr & Eschman, 1970). The armor plate (UMMP 3127) of A. clavatus was collected from the shore of Grand Traverse Bay near Norwood (Dorr & Eschman, 1970). Dunkleosteus is known from a suborbital plate (UMMP 15432) found in a concretion nodule from Squaw Bay, 6.4 kilometers south of Alpena on US 23 (Dorr & Eschman, 1970). D. lafargei is known from an incomplete, disarticulated specimen (CMNH 50215) from Paxton Quarry (Lafarge North America, Inc., Alpena Cement Plant, Great Lakes Region), Alpena (Carr & Jackson, 2005). Invertebrate fossils from this formation include brachiopods and cephalopods (ammonoids), demonstrating relatively low invertebrate diversity compared to Middle Devonian formations (Ehlers & Kesling, 1970; Gutschick & Sandberg, 1991; Hannibal, Carr & Frye, 1992). This is likely reflective of the open water habitat the Antrim Shale was deposited in, a contrast to the reefs and nearshore depositional environments of Michigan’s Middle Devonian localities (Gutschick & Sandberg, 1991). The Antrim Shale is a typical Late Devonian North American black shale, containing large quantities of black mud rich in organic matter from deposition on a poorly oxygenated ocean floor (Dorr & Eschman, 1970; Roen, 1984).

The Environment and Assemblages of Devonian Michigan

During the Middle Devonian, Michigan was located underneath a shallow, tropical sea (Briggs, 1959). Pinnacle reefs were situated in a ring around what is now the Lower Peninsula (Dorr & Eschman, 1970). Most Middle Devonian localities are associated with these reef formations due to abundant invertebrate life and bioherm construction, contributing to rock formation (Ehlers & Kesling, 1970). Much of the center of the Lower Peninsula is covered with a thick layer of glacial till, preventing detailed paleontological study (Lilienthal, 1978). Sites situated along the northern edge of the Lower Peninsula give a fairly good window into the structure of Michigan’s Middle Devonian ecosystems (Ehlers & Kesling, 1970; Dorr & Eschman, 1970). These sites present diverse invertebrate biotas including crinoids, trilobites, cephalopods, gastropods, corals, bryozoans, brachiopods, and blastoids (Pohl, 1930; Ehlers & Kesling, 1970). The vertebrate fauna includes numerous ‘placoderms’ (arthrodires, petalichthyids, and ptyctodonts), chondrichthyans (ctenacanths), lungfish, onychodonts, and ‘acanthodians’ (Dorr & Eschman, 1970).

Vertebrate material, most of it at least partially articulated, from the Devonian of Michigan is usually associated with numerous benthic shelled invertebrates, unlike most sites of similar age outside of North America (Ehlers & Kesling, 1970; J Stack, pers. obs., 2018). This type of assemblage is also found in the similarly-aged Columbus and Delaware Limestones of Ohio (Eastman, 1907; Westgate & Fischer, 1933; Wells, 1944; Denison, 1978; Martin, 2002). In several formations, fishes and invertebrates are found in association within the same conglomerate or slab (Fig. 5D).

Michigan’s faunas exhibited changes in the number, type, and diversity of fossils from the Middle to Late Devonian. Most Late Devonian non-vertebrate fossils are cephalopods, brachiopods, and assorted plant fossils, marking a significant change from the rich invertebrate fauna from the Middle Devonian (Ehlers & Kesling, 1970; Hannibal, Carr & Frye, 1992). In addition, the vertebrate assemblage in the Late Devonian is dominated by arthrodire and ptyctodont ‘placoderms,’ which is a much less diverse fish fauna than what was present in the Middle Devonian (Dorr & Eschman, 1970; J Stack, pers. obs., 2018). As noted above, the Antrim Shale was deposited in an open water pelagic habitat with little to no benthic community (Gutschick & Sandberg, 1991). There was likewise a shift in rock types from primarily limestone in the Middle Devonian to an alternating pattern of shale and limestone in the Late Devonian (Dorr & Eschman, 1970). The Antrim Shale, the only vertebrate-bearing formation from the Late Devonian in Michigan, is constructed in this manner (Dorr & Eschman, 1970; Gutschick & Sandberg, 1991). This alteration suggests that sea levels shifted several times during this interval, which would have contributed to the loss of older reef structures and upwelling of anoxic waters for shale deposition (Roen, 1984; Sandberg, Morrow & Ziegler, 2002). Similar black shale deposition has been recorded from the Devonian of other areas of North America, such as New York, Tennessee, Ohio, and Kentucky (Roen, 1984). In Michigan, coral reefs disappear at the base of the Frasnian with the deposition of black shales (Ehlers & Kesling, 1970). This loss precedes bioherm elimination in other regions of the world, where corals are virtually eliminated by the climate-driven Frasnian-Famennian Kellwasser events (Kiessling, Simpson & Foote, 2010).

Devonian Fishes of Michigan

Here we describe what is known about the occurrence and distribution of major vertebrate clades and lineages from the Devonian of Michigan. Because of the aforementioned absences in the record and new discoveries, much of the information comes from personal observation as noted.

Arthrodires

Protitanichthys rockportensis

Protitanichthys rockportensis is one of the few endemic species of fish from the Middle Devonian of Michigan (Case, 1931; Miles, 1966; Fig. 6). However, P. rockportensis likely belongs to a particularly widespread group of arthrodires, including Dunkleosteus as well as Chinese and Moroccan taxa (Zhu & Zhu, 2013). P. rockportensis is known from 59 specimens housed in the MHM and UMMP that come from the Rockport Quarry Limestone Formation at the abandoned Kelly Island Limestone Quarry at Rockport State Park, Alpena County and one specimen (JS 101) that is known from the Four Mile Dam Formation at the Betsie Bay Rockpiles in Elberta, Benzie County (Dorr & Eschman, 1970, L Sallan, pers. obs., 2018). Most specimens of P. rockportensis consist of disarticulated and incomplete pieces of dermal bone from the head shield (Miles, 1966; Fig. 6C). The poor quality and lack of articulation in these specimens is caused by breakage during quarrying operations, scavenging, and high-energy water flow in the environment of deposition (Ehlers & Kesling, 1970; J Stack, pers. obs., 2018). The majority of the specimens are from fish that would have been about a meter in length in life, but a few rare individuals may have been over two meters in total length, making them very large for coccosteids (Denison, 1978). Most specimens appear to have been adults, but smaller juveniles and possibly senescent animals have been identified (J Stack, pers. obs., 2018). A well-preserved specimen (4M) of the head shield of a small placoderm that resembles Bothriolepis Eichwald, 1840 in size and ornamentation has also been recovered from the Rockport Quarry Limestone at the abandoned Kelly Island Limestone Quarry at Rockport State Park, Alpena County (Fig. 6D). There are also several other small, far less complete specimens from that site bearing similar ornamentation (small, dense tubercles). While these resemble Bothriolepis, they are most likely juvenile specimens of P. rockportensis, as smaller (and presumably younger) P. rockportensis would have had prominent sutures between the plates of their head shield, with their form distinct from larger members of the same species. The relative age of these fishes can be estimated by the distinction of the sutures between the armor plates of the head shield, with older animals bearing less noticeable sutures that fused with age (R Carr, pers. comm., 2012). Better material must be found in order to make a final determination as to these alternative attributions. Because of the relatively high number of specimens, P. rockportensis has been described in detail (Miles, 1966).

Figure 6 Specimens of Protitanichthys rockportensis, an arthrodire that is common in the Middle Devonian sediments of Michigan.

(A) a photograph of an impression of the head of the holotype, from the Rockport Quarry Limestone at the abandoned Kelly Island Limestone Quarry at Rockport State Park, Alpena County, UMMP 12980. Scale bar equals 2 cm. (B) a specimen drawing of UMMP 12980 (redrawn and modified from Case 1931: Fig. 1). Dotted lines represent missing plate boundaries and dashed lines represent sensory grooves. (C) an incomplete nuchal and left paranuchal plate, from the Rockport Quarry Limestone at the abandoned Kelly Island Limestone Quarry at Rockport State Park, Alpena County, UMMP 2981. Scale bar equals 1 cm. (D) a partial headshield, most likely from a juvenile, from the Rockport Quarry Limestone at the abandoned Kelly Island Limestone Quarry at Rockport State Park, Alpena County, 4M, Michigan History Museum. Scale bar equals 1 cm.

Titanicthys

?Titanichthys sp. is known from a single specimen (UMMP 26114) from the Dundee Limestone of Sibley Quarry, Wyandotte (Dorr & Eschman, 1970). Titanichthys is primarily known from open water settings in the Late Devonian, so it is possible that this specimen has been misidentified (Janvier, 2003; Boyle & Ryan, 2017). Indeed, Dorr & Eschman (1970) stated the exact affinity of this specimen was uncertain. Complicating matters further, the specimen is currently missing from museum collections and has never been figured.

Holonema

Two species of Holonema are found in Michigan: H. farrowi and ?H. rugosum (Stevens, 1964; Dorr & Eschman, 1970). Most specimens consist of dorsal and/or ventral shields with distinctive ornamentation of rows and ridges of tubercles, which are easily identifiable even from fragmented remains (Dorr & Eschman, 1970; Fig. 4A). This fish is most common in the Rockport Quarry Limestone at the abandoned Kelly Island Limestone Quarry at Rockport State Park, Alpena County, with 26 specimens registered at UMMP and MHM (Dorr & Eschman, 1970; J Stack, pers. obs., 2018). Two specimens of H. farrowi (UMMP 46647 and UMMP 46648) have been reported from the Newton Creek Limestone at Onaway Stone Quarry, north edge of Onaway, Presque Isle County (Dorr & Eschman, 1970). One specimen of ?H. rugosum (UMMP 3899) has been documented from the Genshaw Formation at French Road, near Long Lake, Alpena County (Dorr & Eschman, 1970). A specimen that Dorr & Eschman (1970) designated as a holonemiid was collected from the Gravel Point Formation of South Point (Gravel Point; UMMP 3129). Finally, another specimen (UMMP 3130) that was designated as a holonemiid by Dorr & Eschman (1970) was collected from an unknown formation in the quarry that is currently known as the Michigan Limestone and Chemical Company quarry in Rogers City, Presque Isle County (Dorr & Eschman, 1970). The flattened body shape and weak bite of Holonema indicates that it was a bottom feeder (Miles, 1971; Denison, 1978).

Dinomylostoma

Dinomylostoma sp. is known from eight specimens (UMMP 3046, UMMP 12974, UMMP 13042, UMMP 13056, UMMP 13148, UMMP 13041, and UMMP 16158) from the Rockport Quarry Limestone at the abandoned Kelly Island Limestone Quarry at Rockport State Park, Alpena County (Dorr & Eschman, 1970). While the reported specimens of this fish are currently missing from museum collections, one of these (the specimen number was not specified) was figured in Dorr & Eschman (1970) and appears to have been accurately identified.

Mylostoma

Mylostoma sp. was previously known in Michigan from one specimen (UMMP 13612) from the Rockport Quarry Limestone at the abandoned Kelly Island Limestone Quarry at Rockport State Park, Alpena County (Dorr & Eschman, 1970). As with many of the other specimens examined by Dorr & Eschman (1970), this specimen was not figured, and cannot be located in the museum collections, even before the recent move of the UMMP collections. In the last few years, J.S. has recovered 39 specimens of small to medium-sized arthrodires that appear to be from Mylostoma sp. from the Four Mile Dam Formation of the Betsie Bay Rockpiles, in Elberta, Benzie County, three specimens from the Alpena Limestone Formation at the Besser Museum Fossil Park in Alpena (BV3, BV6, and BV7), and one specimen from the Rockport Quarry Limestone at the abandoned Kelly Island Limestone Quarry at Rockport State Park, Alpena County (15M). These are now deposited in the MHM as per the permitting procedure for public lands in Michigan, awaiting transfer to the UMMP. The above specimens resemble Mylostoma in having a thick, rounded median dorsal plate and in lacking ornamentation on the surface of the armor plates (Denison, 1978; Fig. 5B). These specimens are usually partially articulated, which is uncommon for placoderm remains from Michigan, and remains of ventral and head shields are also known. However, positive identification as Mylostoma sp. must await the recovery of gnathal plates (Denison, 1978; J Stack, pers. obs., 2018). The size of most of these new specimens suggests that they are most likely juveniles, an inference supported by the coincident collection of a relatively large, poorly preserved specimen (2M) from the Four Mile Dam Formation with similar morphology and lack of tubercles (J Stack, pers. obs., 2018). If this identification is correct, Mylostoma sp. is currently represented by more specimens than any other fish from the Devonian Michigan.

Dunkleosteus

Dunkleosteus sp. is known from one specimen of a partial impression of a suborbital plate (UMMP 15432) from the Antrim Shale of Squaw Bay, 6.4 kilometers south of Alpena, one specimen of a supragnathal (UMMP 16152) from the Alpena Limestone of the Alkali Quarry of Alpena, one specimen of an incomplete anterior ventrolateral plate (UMMP 16156; Fig. 4C) from the Rockport Quarry Limestone at the abandoned Kelly Island Limestone Quarry at Rockport State Park, Alpena County and one specimen (VP.517) from an undetermined site in Alpena (Dorr & Eschman, 1970). These specimens are generally isolated plates, and are not usually articulated. Dorr & Eschman (1970) originally referred to these specimens as Dinichthys sp. Newberry, 1868, but this genus has since been synonymized with Dunkleosteus, with the exception of a single species from the Famennian Huron Shale Member of the Ohio Shale Formation (Carr & Hlavin, 2010). Although these specimens are incomplete, the difference in age between the Michigan specimens and Dinichthys herzeri, along with the resemblance they bear to more complete specimens of Dunkleosteus described by Carr & Hlavin (2010), strongly indicates that they should be attributed to Dunkleosteus sp.

Aspidichthys clavatus

A. clavatus is known from a single specimen (UMMP 3127) from the Antrim Shale of the shore of Grand Traverse Bay near Norwood, Charlevoix County (Dorr & Eschman, 1970). As above, the reported specimen is missing from museum collections and was not figured, so this identification may not be reliable.

Diplognathus lafargei

D. lafargei is known from the Late Devonian of Michigan (Carr & Jackson, 2005). Unfortunately, the recent flooding of Paxton Quarry means that the lens that produced the holotype of D. lafargei is no longer accessible (RL Carr, pers. comm., 2018). D. lafargei is currently known from an incomplete, disarticulated specimen (CMNH 50215) found in a talus slope in Paxton Quarry (Lafarge North America, Inc., Alpena Cement Plant, Great Lakes Region), Alpena (Carr & Jackson, 2005). This specimen includes the suborbital and anterior superognathal plates, along with the right inferognathal, posterior supragnathal, and posterior ventrolateral (Carr & Jackson, 2005).

Trachosteus clarki

?T. clarki is known from a single specimen (UMMP 18206) of a infragnathal plate that is from the Antrim Shale Formation 1.6 kilometers north of Norwood, Charlevoix County (Dorr & Eschman, 1970; Fig. 7). This specimen was designated as uncertain by Dorr & Eschman (1970), most likely because T. clarki is known solely from a disarticulated and incomplete headshield from the Cleveland shale (Denison, 1978). UMMP 18206 resembles the infragnathal of T. clarki in having a low blade and sharp teeth on the biting edge, but there is not enough information available on this fish for this identification to be certain (Denison, 1978).

Figure 7 An inferognathal from the Late Devonian arthrodire ?Trachosteus clarki from the Antrim Shale.

Specimen recovered 1.6 km north of Norwood, MI. UMMP 18206. Scale bar equals 1 cm.

Petalichthyida

Macropetalichthys

?Macropetalichthys sp. is known from a specimen of a spinal and anterior ventrolateral plate (UMMP 14320) from the Middle Devonian of the Dundee Limestone near Trenton (Fig. 3B), a partial headshield (32M; Fig. 8A) from the Four Mile Dam Formation of the Betsie Bay Rockpiles, in Elberta, Benzie County, and from partially articulated plates from the anterior portion of the headshield (BV 4; Fig. 8B) from the Alpena Limestone Formation at the Besser Museum Fossil Park in Alpena (Dorr & Eschman, 1970; J Stack, pers. obs., 2018). UMMP 14320 was identified by Dorr & Eschman (1970) as Arctolepis sp. Eastman, 1908 based on its elongated spinal plates ornamented with small spines. However, this specimen (Fig. 3B; UMMP 14320) much more closely resembles Macropetalichthys because the spinal plate is not as recurved as it is Arctolepis, the spines on this spinal plate are more numerous and tightly spaced than those in Arctolepis, and because its spines are present on the outer edge of the spinal plate (unlike Arctolepis, where the spines are present on the interior edge of the spinal plate) (Denison, 1978; Janvier, 2003; J Stack, pers. obs., 2018). Unfortunately, the specimen does not retain any of the diagnostic features of the genus, so we can only tentatively reattribute it (Eastman, 1907; Denison, 1978). Whatever the case, Macropetalichthys is already known from several localities in North America, including the Delaware and Columbus Limestones of Ohio, which were closely associated with Michigan during the Middle Devonian (Eastman, 1907; Denison, 1978; Martin, 2002). Arctolepis, however, is otherwise restricted to the Early Devonian of Spitsbergen (Denison, 1978). The anatomical, temporal, and geographical evidence therefore indicates that UMMP 14320 is far more likely to be from Macropetalichthys than Arctolepis (J Stack, pers. obs., 2018). The rounded anterior portion of the indentation of the headshield in 32M strongly resembles the narrow, rounded rostral plate in the headshield of Macropetalichthys in shape (Eastman, 1897b; Denison, 1978; Fig. 8A). This specimen also resembles Macropetalichthys in bearing sparse, irregularly arranged tubercles (Denison, 1978; Martin, 2002; Fig. 8A). Similarly, the shape of the plates present in BV 4 also strongly resembles the anterior portion of the headshield of Macropetalichthys (Eastman, 1897b; Denison, 1978; Martin, 2002; Fig. 8B). In particular, one of the plates (labeled R in Fig. 8B) looks very similar to the rounded, blunt rostral plate seen in Macropetalichthys (Denison, 1978). Despite the strong similarity in shape between these specimens and Macropetalichthys, neither of these specimens are complete enough to make their attribution to Macropetalichthys definitive.

Figure 8 Newly discovered specimens of ?Macropetalichthys sp.

(A) a partially complete headshield from ?Macropetalichthys sp. Specimen found in the Four Mile Dam Formation of the Betsie Bay Rockpiles, Elberta, Benzie County. Abbreviations: Hd, Head Shield (incomplete); Un, Unidentified Armor Plate; R, Rostral Plate? Scale bar equals 2 cm. 32M, Michigan History Museum. (B) crushed and partially articulated pieces of armor from the anterior portion of the headshield of ?Macropetalichthys sp. Specimen found in the Alpena Limestone Formation at the Besser Museum Fossil Park, Alpena. BV 4, Michigan History Museum. Abbreviations: PrO, Preorbital; R, Rostral. Scale bar equals 1 cm.

Ptyctodontida

Ptyctodus

Specimens of Ptyctodus sp., consisting of isolated tooth plates, have been found in the Dundee Limestone near Trenton, Wayne County (UMMP 14321), the Bell Shale of Rogers City, Presque Isle County (UMMP 14460), the Thunder Bay Limestone of the bluffs on the northeast shore of Partridge Point, 6.4 kilometers south of Alpena, Alpena County (UMMP 3023), the Rockport Quarry Limestone at the abandoned Kelly Island Limestone Quarry at Rockport State Park, Alpena County (UMMP 13045), the Alpena Limestone of Alkali Quarry, Alpena (UMMP 16157), the Potter Farm Formation (uncertain) of “old Wamer’s Brickyard” southwest of Alpena (UMMP 21718), and the Potter Farm Formation of the west edge of Alpena Cemetery, Alpena (UMMP 21817; Dorr & Eschman, 1970). Other single specimens of Ptyctodus are recorded from an unknown formation in Afton Quarry, Cheboygan County (VP.489) and the Traverse Group (unknown formation) of Emmet County (UMMP 14712) (Dorr & Eschman, 1970). While many of these specimens have gone missing since 1970, those figured (specimen numbers not specified) in Dorr & Eschman (1970) suggest their attribution is accurate (J Stack, pers. obs., 2018). There are also reports of other ptyctodont remains, including isolated gnathal plates and articulated specimens, from the Antrim Shale (Elliott et al., 2000). However, it is not clear if the specimens this report is based upon are deposited in museum collections, so they cannot currently be verified. The widespread distribution of Ptyctodus fossils may be due to both the higher preservation potential of hard tooth plates and/or association with abundant shelly invertebrates. Relatively poor taxonomic knowledge of Ptyctodus, a wastebin taxon widely applied to various ptyctodont teeth, may also be a contributing factor (Denison, 1978). In general, ptyctodont tooth plates are common, but articulated remains are not (Denison, 1978; Trinajstic & Long, 2009). Therefore, tooth plates form the basis for the majority of ptyctodontid taxa, including Ptyctodus (Denison, 1978; Trinajstic & Long, 2009). Further taxonomic work on existing specimens, and on North American ptyctodonts in general, is required to determine if the specimens from Michigan all originate from the same genus.

Figure 9 An incomplete armor plate from Eczematolepis sp.

Specimen from the Genshaw Formation. Found near Posen, MI. UMMP 4169. Scale bar equals 1 cm.

Eczematolepis

?Eczematolepis sp. is known from a single partial armor plate (UMMP 4169) from the Genshaw formation, near Posen, Presque Isle County (Dorr & Eschman, 1970; Fig. 9). UMMP 4169 was identified as Bothriolepis sp. by Dorr & Eschman (1970), however we attribute this plate to ?Eczematolepis sp. because its ornamentation of crowded, irregularly arranged, tubercles more closely resembles Eczematolepis (Denison, 1978; Martin, 2002; J Long, pers. comm., 2018). Also, the shape of this plate does not resemble what is seen in Bothriolepis (J Long, pers. comm., 2018). This specimen is not complete enough to make an exact identification, so we designate it as uncertain. Besides this specimen, Dorr & Eschman (1970) identified a large supragnathal plate (UMMP 14374; Fig. 10) from an unknown formation in the Traverse Group of a locality in Alpena, Alpena County as Eczematolepis sp.. The locality this specimen was recovered from is referred to by Dorr & Eschman (1970) as “Locality 650 of the Winchell Survey”, but no other information is available on its geological context. While this specimen appears to the supragnathal of a ptyctodont, Eczematolepis is known only from head or body plates (Denison, 1978). Furthermore, there is uncertainty surrounding whether or not Eczematolepis is a ptyctodont (Denison, 1978; J Long, pers. comm., 2018). Therefore, although this specimen can be concretely identified as a ptyctodont, it cannot be attributed to Eczematolepis. A general lack of taxonomic information for ptyctodonts from the Eifelian of North America means that this specimen cannot be identified at a finer taxonomic level at this point in time (Martin, 2002).

Figure 10 A supragnathal plate from an unknown ptyctodont from the Traverse Group.

Specimen recovered from an unknown locality referred to as “Locality 650 of the Winchell Survey”, in Alpena, Alpena County, MI. UMMP 14374. Scale bar equals 1 cm.

‘Acanthodii’

Gyracanthus

Gyracanthus sp. is known from one specimen (UMMP 1329) from the Gravel Point Formation of South Point (Gravel Point), Little Traverse Bay, Charlevoix County (Dorr & Eschman, 1970). This specimen is currently missing from the UMMP and was not figured by Dorr & Eschman (1970). This spine-based identification is therefore not verifiable, particularly as Devonian specimens of this widespread Carboniferous genus are dubious and in need of re-examination (Turner, Burrow & Warren, 2005).

Machaeracanthus

Machaeracanthus sp. is reported from one specimen (UMMP 3521) from the Dundee Limestone of Monroe County, two specimens (UMMP 26111 and UMMP 26112) from the Dundee Limestone of Sibley Quarry, Wyandotte, Wayne County, one specimen (UMMP 13047) of uncertain status from the Rockport Quarry Limestone at the abandoned Kelly Island Limestone Quarry at Rockport State Park, Alpena County, and two specimens (UMMP 47691 and UMMP 47692) from the Newton Creek Limestone at Onaway Stone Quarry, Presque Isle County (Dorr & Eschman, 1970, J Stack, pers. obs., 2018; Fig. 11). The specimen from the Rockport Quarry Limestone (UMMP 13047) was identified as A. gracillimus by Dorr & Eschman (1970), but an examination of the specimen demonstrated significant differences in the structure of this spine compared to what is known from A. gracillimus (Maisey, 1983). This spine is long and thick with a smooth surface, and therefore much more closely resembles the spines of Machaeracanthus (Denison, 1979; Maisey, 1983; Fig. 11B). As shown by the specimen list (available in the supplemental files), Machaeracanthus is relatively common in the Middle Devonian of Michigan and closely associated areas (Eastman, 1907; Wells, 1944; Dorr & Eschman, 1970; Denison, 1978). In contrast, A. gracillimus is known only from the Carboniferous of Iowa in North America, after a major mass extinction event (Wellburn, 1901; Zangerl, 1981; Maisey, 1983; Itano, Houck & Lockley, 2003; Elliott et al., 2004; Brusatte, 2007; Sallan & Coates, 2010).

Figure 11 Specimens of the ‘acanthodian’ Machaeracanthus sp.

(A) a large spine from Machaeracanthus sp., from the Dundee Limestone of London Township, Monroe County, UMMP 3521. Scale bar equals 1 cm. (B) a spine from ?Machaeracanthus sp. (originally identified as A.gracillimus), from the Rockport Quarry Limestone at the abandoned Kelly Island Limestone Quarry at Rockport State Park, Alpena County, UMMP 13047. Scale bar equals 1 cm.

Oracanthus

Oracanthus sp. is known from a fin spine (UMMP 23495) from the Norway Point Formation of the Four Mile Dam, about 5.6 km northwest of Alpena, Alpena County (Dorr & Eschman, 1970). As above, this specimen was unfigured by Dorr & Eschman (1970) and it is missing, so we cannot verify its identity.

Chondrichthyes

Acondylacanthus gracillimus

A fin spine specimen of A. gracillimus (UMMP 26523) was collected from the Dundee Limestone of Sibley Quarry, Wyandotte, Wayne County, (Dorr & Eschman, 1970). This specimen is now missing from the UMMP, so we cannot determine if this identification is reliable or if it was misidentified in the same way as UMMP 13047. This would be the earliest reported specimen of Acondylacanthus by far; other occurrences are clustered in the Carboniferous of the US and the UK (Wellburn, 1901; Maisey, 1983; Itano, Houck & Lockley, 2003; Elliott et al., 2004; Brusatte, 2007).

Tamiobatis

?Tamiobatis sp. is a small chondrichthyan reported from one specimen (UMMP 13147) of a fin spine from the Rockport Quarry Limestone at the abandoned Kelly Island Limestone Quarry at Rockport State Park, Alpena County (Dorr & Eschman, 1970; Fig. 4D). Dorr & Eschman (1970) identified this spine as Ctenacanthus Agassiz 1835, but comparisons of this specimen with more recent descriptive work disputes this attribution (Maisey, 1982; Williams, 1998). This specimen more closely resembles a fin spine impression from Tambiobatis from the Cleveland Shale (Williams, 1998). More complete material from Michigan is needed to make a concrete diagnosis, so this assignment is designated as uncertain.

Onychodontiformes

Onychodus

Onychodus is found in several parts of Michigan’s geological column (Dorr & Eschman, 1970), including the Dundee Limestone of London Township, Monroe County (UMMP 22006; Fig. 3A), Sibley Quarry, Wyandotte, Wayne County (UMMP 26113) as well as an uncertain specimen from the Gravel Point Formation of the shore of the Little Traverse Bay, Charlevoix County (UMMP 14370) (Dorr & Eschman, 1970). UMMP 22006 was identified to the species level, O. sigmoides (Dorr & Eschman, 1970). In most cases, the genus is represented solely by its large distinctive tooth whorls, with the exception of one lower jaw (UMMP 26113; Dorr & Eschman, 1970).

Dipnoi

Chirodipterus onawayensis

C. onawayensis is the only lungfish known from the Devonian of Michigan, and thus far is represented by a single specimen which was preserved well enough to allow diagnosis as a new species (Schultze, 1982; Long, 1995). The holotype of C. onawayensis represents the left side of the skull and jaws, and was collected from the Onaway Stone Quarry, which is north of Onaway in Presque Isle County (Schultze, 1982; Fig. 12). This specimen was unnumbered at the GLAPM and appears to be missing (Schultze, 1982). It is similar to Chirodipterus australis Miles, 1977 from Gogo in Australia, and possesses the powerful jaws typical of a durophagous Devonian lungfish (Schultze, 1982; Long, 2011).

Figure 12 The skull of Chirodipterus onawayensis.

Specimen recovered from the Newton Creek Limestone of Onaway Stone Quarry, north of Onaway, Presque Isle County. Specimen photo from Schultze (1982), Fig. 2, modified and reprinted with permission of Taylor and Francis Ltd, http://www.tandfonline.com. This specimen is reported to be unnumbered at the Great Lakes Area Paleontological Museum, but is missing. Scale bar equals 1 cm.

Discussion

Despite proximity to major research institutions and collections, the rich reef and nearshore faunas of the Middle Devonian of North America have been neglected in recent decades, particularly relative to similarly-aged localities in even more remote areas of Antarctica, Australia, and Morocco (Gardiner, 1984; Derycke, Cloutier & Candilier, 1995; Blieck & Leliévre, 1995; Elliott et al., 2000; Janvier, 2003; Rücklin, 2010; Sallan & Coates, 2010; Friedman & Sallan, 2012). There are significant gaps in the total Devonian record in Michigan, with vertebrates in some intervals, particularly the Late Devonian, poorly represented and deficient in number relative to similarly-aged horizons in Ohio (Dorr & Eschman, 1970; Carr & Jackson, 2008). Complicating matters, a large proportion of previously published and catalogued specimens could not be located in the paleontological collections at the University of Michigan, leaving only brief and incomplete documentation as proof of their existence (Dorr & Eschman, 1970). In addition, a large number of more recently recovered specimens are resident in amateur collections - the result of a lack of professional efforts in the state in recent decades - and cannot be used for scientific purposes.

Never-the-less, examination of available new and old material shows that Michigan is much richer in diversity and sheer number of fish specimens than previously thought. This has revealed several previously unreported but likely significant biogeographical and diversity patterns, including a shift in environment and faunas between the Middle and Late Devonian and greater connections to nearby basins. In addition, new localities have produced co-occurring, well-preserved articulated vertebrate and invertebrate material, a rarity in the Paleozoic record outside of Michigan.

Figure 13 A representation of the Devonian vertebrate fauna known from Michigan.

Animals not to scale. Drawing by L.S. (A) The vertebrate fauna from the Middle Devonian of Michigan. (1) Acondylacanthus (Chondrichthyes); (2), Dinomylostoma (Arthrodira; ‘Placodermi’); (3), Chirodipterus (Dipnoi; Sarcopterygii); (4), Dunkleosteus (Arthrodira; ‘Placodermi’); (5), Onychodus (Onychodontida; Sarcopterygii); (6), Mylostoma (Arthrodira; ‘Placodermi’); (7), Protitianichthys (Arthrodira; ‘Placodermi’); (8), Oracanthus (Acanthodida; ‘Acanthodii’); (9), Machaeracanthus (Ischnacanthida; ‘Acanthodii’); (10), Gyracanthus (Gyracanthida; ‘Acanthodii’); (11), Eczematolepis (Ptyctodontida; ‘Placodermi’); (12), Holonema (Arthrodira; ‘Placodermi’). (B) The vertebrate fauna from the Late Devonian of Michigan. (13) Aspidichthys (Arthrodira; ‘Placodermi’); (14), Trachosteus (Arthrodira: ‘Placodermi’); (15), Diplognathus (Arthrodira; ‘Placodermi’); (16), Ptyctodontida indet. (‘Placodermi’).

There is a definite shift in fish diversity, geologic range, and number between the Middle and Late Devonian deposits of Michigan. Placoderms, acanthodians, dipnoans, onychodonts and sharks are found in fair numbers in the primarily nearshore settings of the Middle Devonian (Dorr & Eschman, 1970; Fig. 13A). This record includes 16 confirmed genera of fishes from 201 reported specimens sourced from 11 separate formations (Dorr & Eschman, 1970; J Stack, pers. obs., 2018). In contrast, fish fossils from Late Devonian pelagic settings come from just four confirmed genera, all arthrodiran or ptyctodont placoderms, in the Antrim Shale formation (Dorr & Eschman, 1970; Carr & Jackson, 2005; Fig. 13B). Reports from Elliott et al. (2000) indicates that there may be a greater diversity of vertebrates present in the Antrim Shale than what is represented in museum collections (Dorr & Eschman, 1970; J Stack, pers. obs., 2018). Even if these reports are confirmed it is still considerably less diverse and abundant than what is observed in Middle Devonian deposits. A similar change is seen in the invertebrate fauna; a thriving reef and nearshore fauna hosting a multitude of life, including crinoids, trilobites, bryozoans, corals, blastoids, brachiopods, cephalopods, gastropods and stromatoporoids in the Middle Devonian is succeeded by scattered fossils of brachiopods and cephalopods further offshore in the Late Devonian (Dorr & Eschman, 1970; Ehlers & Kesling, 1970; Hannibal, Carr & Frye, 1992; R Carr, pers. comm., 2014).

The contrast between the Middle and Late Devonian vertebrate and invertebrate faunas in Michigan is due to differences in collection intensity, rock exposure, and environmental representation. There are at least a dozen well-documented Middle Devonian localities from Michigan that have been the focus of both professional and amateur collectors (Dorr & Eschman, 1970; J Stack, pers. obs., 2018). These localities preserve a wide variety of habitats (mostly near-shore, reef habitats) and have a large amount of exposed rock (especially in limestone quarries) (Ehlers & Kesling, 1970; J Stack, pers. obs., 2018). In contrast, the Late Devonian of Michigan is represented by four localities from a single, black shale heavy formation that have comparatively little rock exposed (Ehlers & Kesling, 1970). Furthermore, Ehlers & Kesling (1970) argued that an abundant vertebrate fauna is unlikely to be recovered from the Antrim Shale because of the hardness of the concretions from this formation and the rarity of vertebrate specimens within them. It is therefore unlikely that the small amount of attention Late Devonian localities have received from amateur collectors is the driving factor behind the observed drop in the diversity and abundance of vertebrates. It is notable that the Antrim Shale was deposited in an open water pelagic habitat with little to no benthic community (Gutschick & Sandberg, 1991). None of the localities from the Middle Devonian of Michigan preserve this kind of habitat (Ehlers & Kesling, 1970). This difference in environment between the Middle and Late Devonian deposits is most likely a major factor contributing to the observed shift in the diversity of the vertebrate and invertebrate faunas.

Re-examination of Michigan’s Devonian fossils sheds some light on biogeographic and dispersal patterns for North American fishes of this age. A complete absence of endemic taxa at the genus level within Michigan suggests that there were few barriers to dispersal with other parts of the mid-continental region of the Old Red Sandstone Continent during the Middle Devonian (Newberry, 1889; Dorr & Eschman, 1970; Denison, 1978; Markus, 1998; Palmer, 1999; Warren et al., 2000; Elliott et al., 2000; Thomson & Thomas, 2001; Sepkoski, 2002; Johanson et al., 2007; Carr & Jackson, 2008; Carr & Hlavin, 2010). However, it is possible that the aforementioned lack of taxonomic work and collection effort has resulted in the incorrect attribution of distinct species from Michigan to taxa from the wider region. Regardless, the types of fish found in the Devonian sediments of Michigan are fairly typical for the eastern United States (Newberry, 1873; Cluff, 1980).

Michigan’s fish fauna shares characteristics with several similarly-aged faunas from the Middle Devonian of North America. Michigan’s Middle Devonian vertebrate fauna is closest in composition to the similarly aged Delaware and Columbus Limestones of central Ohio, with which it shares many taxa, including Machaeracanthus, Gyracanthus, Holonema, Macropetalichthys, Protitanichthys, Onychodus, Dunkleosteus, Ptyctodus, and Eczematolepis (Eastman, 1907; Westgate & Fischer, 1933; Wells, 1944; Dorr & Eschman, 1970; Denison, 1978; Martin, 2002). This suggests that the parts of Michigan and Ohio that these deposits represent were closely connected during this period of time, yet the preservational mode was quite different. Many of the described fish remains from the Delaware and Columbus Limestones are very small and worn, concentrated into bone beds where vertebrate remains are more common than macroscopic invertebrate fossils (Westgate & Fischer, 1933; Wells, 1944). This is very different than Michigan, where fish remains are generally large to medium size pieces of armor or spines that are usually unworn (J Stack, pers. obs., 2018). However, Martin (2002) describes the remains of more complete specimens of placoderms (petalichthyids and ptyctodonts) and onychodonts from other, lesser known sections of the Delaware and Columbus Limestones, indicating that some beds are more similar to Michigan in preservation and assemblage composition.

The Middle Devonian fish fauna of Michigan is also similar to the vertebrate fauna known from the Onondaga Limestone of New York (Eifelian, Upper Ulsterian), which has a similar environment to and is comparable in age to the Dundee Limestone (Brett & Ver Straeten, 1994; Brett et al., 2011). Indeed, the Onondaga Limestone shares all but one of the taxa found in the Dundee Limestone, including Ptyctodus, Machaeracanthus, Onychodus, Eczematolepis, and Macropetalichthys (Eastman, 1907; Dorr & Eschman, 1970; Denison, 1978). A larger number of vertebrate taxa have been reported from the Onondaga Limestone, although this might be an artifact of the lesser number of outcrops of this age in Michigan and a lack of collecting effort at said outcrops, rather than reflective of real differences in diversity (Eastman, 1907; Dorr & Eschman, 1970; Denison, 1978).

The correlation between the vertebrate faunas of Michigan and New York continues into the Givetian (Erian). The rocks of the Traverse Group in Michigan and the Hamilton Group of New York are similar in age and share two vertebrate taxa, Machaeracanthus and Dunkleosteus (Eastman, 1907; Dorr & Eschman, 1970; Denison, 1978; Brett & Ver Straeten, 1994). This is despite an environmental shift that caused major changes in sedimentation, paleoecology, faunas, and basin geometry that occurred in the transition between the Onondaga Limestone and the Hamilton Group (Ver Straeten, Griffing & Brett, 1994). This shift had a major effect on the invertebrate fauna of the region, causing extinctions of some of the endemic Onondaga faunas (Ver Straeten, Griffing & Brett, 1994). While it is not clear what effect this shift had on the vertebrate fauna of New York, it is evident that a close connection between the fish faunas of Michigan and New York continued from the Eifelian (Ulsterian) into the Givetian (Erian).

The documented loss in the amount and diversity of fossil material in the Late Devonian of Michigan makes detailed comparison with other Late Devonian faunas difficult. However, the vertebrate genera found in the Late Devonian of Michigan, D. lafargei, ?T. clarki, ptyctodonts, A. clavatus, and Dunkleosteus, are also found in open ocean sediments of the Late Devonian Cleveland Shale (Newberry, 1889; Winston and Walker, 1956; Dorr & Eschman, 1970; Denison, 1978; Carr & Jackson, 2008). The Cleveland Shale has been the focus of intense collecting efforts for the past 150 years and has outcrops both within a major metropolitan area and on the path of a major highway, while very little collecting has been conducted in the relatively remote, low abundance, and difficult-to-sample Antrim Shale (Hlavin, 1976). In contrast, the Cleveland Member of the Ohio Shale is a Konservat-Lagerstätten, and is considered one of the most diverse vertebrate faunas from the Devonian (Carr & Jackson, 2008). Therefore, the gap in the diversity and number of fish specimens between the Late Devonian of Michigan and the Late Devonian of Ohio is probably largely the result of the differences in preservation between these sites, along with a lack of organized collection effort in Michigan’s Late Devonian sediments by both professionals and amateurs.

A notable occurrence, or non-occurrence, in the Middle Devonian fish fauna of Michigan is a complete lack of antiarch placoderms (Dorr & Eschman, 1970). Additional benthic, nearshore forms, such as gyracanthids and ptyctodonts, are also poorly represented relative to other kinds of fishes (such as arthrodires) in Michigan’s sediments. The relative absence of benthic-associated fishes contrasts greatly with the large amount of benthic invertebrate material at vertebrate-bearing localities, which indicates that preservation of the sea floor is not the issue. It is possible that the rarity of antiarchs is purely the product of a lack of collection effort outside of a handful of sites. However, it appears that antiarchs are also uncommon in other Middle Devonian sites that are closely related to deposits of the same age in Michigan (Eastman, 1907; Westgate & Fischer, 1933; Wells, 1944). This is despite the fact that antiarchs have been recovered from nearshore marine and estuarine settings elsewhere, such as the famous marine tetrapod assemblage, Andryevka-2 (Sallan & Coates, 2010; Friedman & Sallan, 2012).

Another interesting aspect of the vertebrate record from the Middle Devonian of Michigan is the occurrence of partially articulated vertebrate material preserved alongside invertebrate remains not only in the same formations, but in the same rocks (Fig. 5D). This pattern is consistent in several separate formations and sites. It is rare to find articulated fish remains, rather than ichthyoliths like teeth, directly associated with complete invertebrate remains, especially articulated crinoids, in the Middle Paleozoic (L Sallan, pers. obs., 2018; Sallan et al., 2011). This direct association can be used to concretely determine which invertebrate taxa lived directly alongside vertebrates, potentially shedding light on the interactions and associations between these groups.

Much more fieldwork is required to fully understand the Devonian vertebrate fauna from Michigan. Recent efforts have revealed a surprising number of new occurrences of fishes in geological formations where they were previously considered absent. P. rockportensis was once thought to be restricted to the Rockport Quarry Limestone, but has now been documented from the Four Mile Dam Formation and possibly the Alpena Limestone Formation (Dorr & Eschman, 1970; J Stack, pers. obs., 2018). Mylostoma sp. was previously only known from an isolated specimen (UMMP 13612) from the Rockport Quarry Limestone, but it is now also known from large numbers of recently collected specimens from the Four Mile Dam Formation and several specimens (BV3, BV6, and BV7) from the Alpena Limestone Formation (Dorr & Eschman, 1970). Additionally, fish fossils had previously never been documented from the Four Mile Dam Formation (J Stack, pers. obs., 2018; Dorr & Eschman, 1970). These findings, which are a result of intensified collecting from a handful of the vertebrate-bearing Middle Devonian localities in Michigan, show that these long-neglected localities are still productive. Further collecting at sites that have been ignored for decades will almost certainly lead to more discoveries. Renewed search efforts will create a less biased understanding of the Late Devonian fish fauna of Michigan, allowing more accurate comparisons to other Late Devonian faunas to be made and the ecology and biogeography of Devonian marine fishes to be more completely known.

Conclusions

Novel information about the ecology, diversity, and number of the fishes from the Devonian of Michigan has been revealed by new surveys of old material and from new specimens obtained through recent collecting efforts. These include many previously unrecognized patterns, such as dramatic losses in vertebrate diversity between the Middle and Late Devonian that are likely due to the differences in rock exposure and environmental representation between these time periods. We have also documented strong connections with other North American pelagic faunas, and the exceptional occurrence of partially-articulated fishes preserved alongside benthic invertebrates. These discoveries show that there is still a lot of work to be done in Michigan’s vertebrate-bearing Devonian sediments, with implications for our understanding of Devonian fish faunas as a whole.

Supplemental Information

Table S1 Vertebrate bearing localities from Dundee Limestone, Bell Shale, and Rockport Quarry Limestone

Click here for additional data file.

Table S2 Vertebrate bearing localities from the Genshaw Formation, Newton Creek Limestone, Gravel Point Formation, and Alpena Limestone

Click here for additional data file.

Table S3 Vertebrate bearing localities from the Alpena Limestone, Four Mile Dam Formation, Norway Point Formation, and Potter Farm Formation

Click here for additional data file.

Table S4 Vertebrate bearing localities from the Thunder Bay Limestone and the Antrim Shale

Click here for additional data file.

Supplemental Information 1 Faunal lists for Devonian fish localities from Michigan

Click here for additional data file.

Supplemental Information 2 Examined specimens in Michigan museum collections

Click here for additional data file.

We would like to thank Dr. Michael Gottfried and Laura Abraczinskas for providing crucial data from Michigan State University’s fossil collection. We would also like to thank Dr. Robert Carr, for helping to identify fossils, providing information on his work in the Antrim Shale, and providing insightful reviews on this paper. We would also like to thank Dr. John Long for his comments, suggestions, and providing access to his collection of placoderm literature, which were all crucial in the completion of this paper. We would also like to thank Dr. Jeffrey Wilson for his valuable comments on the manuscript, Dr. Adam Rountrey, for providing access to the University of Michigan’s fossil fish collection, Bruce Tobin, for donating crucial specimens used in this study, John Paul Hodnett, for providing key insights on shark specimens, Scott Peters, for helping with the donation of the specimens collected for this study, and Joseph Kchodl, who has been instrumental in pin-pointing the geological affinities of many of the fish fossils and localities used in the data collection for this study. Finally, we would like to thank Dr. Murray Borrello for his encouragement and support.

Additional Information and Declarations

Competing Interests

Author Contributions

Field Study Permissions

Data Availability

The authors declare there are no competing interests.

Jack Stack performed the experiments, analyzed the data, prepared figures and/or tables, authored or reviewed drafts of the paper, approved the final draft, and carried all fieldwork and museum surveys.

Lauren Sallan conceived and designed the experiments, prepared figures and/or tables, authored or reviewed drafts of the paper, approved the final draft.

The following information was supplied relating to field study approvals (i.e., approving body and any reference numbers):

Fieldwork on public lands was approved by the State of Michigan Department of Natural Resources and State Historic Preservation Office (Permit AE2016-10).

The following information was supplied regarding data availability:

The raw data are provided in the Supplemental Files.

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
