# Peer review of "An examination of the Devonian fishes of Michigan"

_PeerJ, doi:10.7717/peerj.5636_

## Round 0.1 · original submission · Minor Revisions

This manuscript requires only minor revisions before it can be considered for acceptance for publication.

Please note the additional files provided by both reviewers.

·

Basic reporting

Minor structure comment: references to figure are typically in the format of Author (1999: fig. 1). I noted this in the text as a PDF sticky note.

Figure captions easier to read if each letter is in bold. One figure uses A) vs. A, in other figures.

Experimental design

no comment

Validity of the findings

no comments

Additional comments

I have made all my comments as PDF sticky notes. I have a question on two figures whether they are in focus or not.

I have a question as to how the supplemental data is presented. The excel table far exceeds a normal page but if presented as an excel file then the reader can view it.

If you add indet. aspinothoracid arthrodires to the text based on the Antrim list in Elliott et al., 2000 then you need to add it also to the supplemental list.

My institution is listed as Columbia Univeristy Chicago -- it should be Concordia University Chicago

·

Basic reporting

The paper is well written and for the most part well illustrated, although one images I've noted in the review copy could be improved. It reviews the fossil fish faunas well, but could include a few more explanatory comments about dubious genera (eg Ptyctodus, ,a s noted in my comments on the review pdf). Data well presented, structure of the paper is good.

Experimental design

All OK, not really relevant to palaeontology review papers like this.

Validity of the findings

Mostly good, with in depth discussion of depositional environment setc.
I've noted a few things I disagree with or take issue with in my comments on the text.

1. Bothriolepis in Fig 9 is NOT Bothriolepis but an Arthrodire or Petalichthyid style of plate. Its certainly not an AVL of an antiarch as it lacks evidence of the lateral lamina or brachial process, and is way too narrow.

2. Fig 10 supragnathal plate can't be attributed yet to Eczematolepis as no-one has yet proven its a ptyctodontid. I have my own evidence for this (not yet published though, , as I've studied this material in the Smithsonian and Field Museum collections).

Additional comments

It's a good paper with fine illustrations of the faunas, but needs a few more points discussed as I've drawn attention to in the comments attached to the review pdf. Some specific points to address:-

1. Bothriolepis in Fig 9 is NOT Bothriolepis but an Arthrodire or Petalichthyid style of plate. Its certainly not an AVL of an antiarch as it lacks evidence of the lateral lamina or brachial process, and is way too narrow.

2. Fig 10 Supragnathal plate can't be attributed yet to Eczematolepis as no-one has yet proven its a ptyctodontid. I have my own separate evidence for this (not yet published though, as I've studied this material in the Smithsonian and Field Museum collections) - perhaps something I could contact the authors about separately as I would welcome a future collaboration on the project I've started with this genus.

---

## Round 0.2 · Minor Revisions

While the manuscript is good in technical terms of content the bibliography was carelessly assembled. Capitalization is often incorrect and taxonomic names, book titles, and journal titles are not italicized or complete. Please check each item either against the original publication or using the Bibbliography of Fossil Vertebrates.These points must be addressed before the manuscript can be accepted for publication.

---

## Round 0.3 · accepted · Accept

The revised version of the manuscript is acceptable for publication in PeerJ.

#